# Measurement of Methane and Ammonia Emissions from Compost-Bedded Pack Systems in Dairy Barns: Tilling Effect and Seasonal Variations

**DOI:** 10.3390/ani13111871

**Published:** 2023-06-04

**Authors:** Esperanza Fuertes, Joaquim Balcells, Jordi Maynegre, Gabriel de la Fuente, Laura Sarri, Ahmad Reza Seradj

**Affiliations:** Department of Animal Science, University of Lleida, Alcalde Rovira Roure 191, 25198 Lleida, Spain; esperanza.fuertes@udl.cat (E.F.);

**Keywords:** dairy cattle, manure, greenhouse gas, ammonia, housing system, compost-bedded pack

## Abstract

**Simple Summary:**

Understanding contaminant gas emissions from manure management systems such as compost-bedded pack systems, whose popularity keeps increasing among dairy housing systems, is a necessary tool when it comes to evaluating their environmental impact. This work showed that CH_4_ and NH_3_ emissions coming from this system should not be underestimated, especially during the warmer months of the year. As emissions coming from manure in compost-bedded pack systems have not been extensively studied yet, we found that the composting process occurring daily on manure from compost-bedded pack barns leads to great amounts of CH_4_ and NH_3_ emissions. This is why despite the potential benefits to animal health and welfare, contaminant gases originating from manure from compost-bedded pack systems should be taken into account.

**Abstract:**

Dairy cattle contribute to environmental harm as a source of polluting gas emissions, mainly of enteric origin, but also from manure management, which varies among housing systems. Compost-bedded pack systems use manure as bedding material, which is composted in situ daily. As current literature referring to their impact on NH_3_ and CH_4_ emissions is scarce, this study aims to characterize the emissions of these two gases originating from three barns of this system, differentiating between two emission phases: static emission and dynamic emission. In addition, the experiment differentiated emissions between winter and summer. Dynamic emission, corresponding to the time of the day when the bed is being composted, increased over 3 and 60 times the static emission of NH_3_ and CH_4_, respectively. In terms of absolute emissions, both gases presented higher emissions during summer (1.86 to 4.08 g NH_3_ m^−2^ day^−1^ and 1.0 to 4.75 g CH_4_ m^−2^ day^−1^ for winter and summer, respectively). In this way, contaminant gases produced during the tilling process of the manure, especially during the warmer periods of the year, need to be taken into account as they work as a significant factor in emissions derived from compost-bedded pack systems.

## 1. Introduction

Livestock contribute around 14.5% to anthropogenic greenhouse gas (GHG) emissions [1]; this fact, added to the expected increase in demand for livestock products, mostly in developing countries [2], makes this sector play a crucial role in climate change [3]. Emissions of contaminant gases in dairy systems mainly come from ruminal fermentation (i.e., CH_4_), but a significant fraction results from manure management and storage. Up to 10% of GHG emissions originate from this activity, and it is also expected to keep increasing over the next years [1,4]. Furthermore, it must be taken into account that circa 60% of anthropogenic NH_3_ emissions in Europe come from manure management [5], so its control is also of great importance.

Available GHG and NH_3_ emission data are based on estimation approaches including experimental measurements and system modelling for pig, poultry, and cattle production [6,7,8]. However, information and data on emissions from open, naturally ventilated dairy cattle barns are still needed to establish reliable emission models. In this sense, previous works from this group [9,10] demonstrated how different floor types and manure handling methods are crucial for controlling and minimizing contaminant gases’ emission.

Recently, compost-bedded pack (CBP) barn systems have received increasing attention. This system consists of an open resting area (between 20 and 30 m^2^ per cow) where cows lie over their own manure, which daily composted daily “in situ” by the tillage of a rotary harrow or cultivator [11]. An alternative stocking rate density may require less space (minimum of 15 m^2^) when feed alleys are daily scraped and the resultant slurry is removed and stored in a pile [12]. Because of the low cost and positive effects on animal welfare, health, and milk quality, this system has become an alternative to the loose-housing systems based on cubicles [13]. Nevertheless, its environmental impact on contaminant gas emissions, such as CH_4_ and NH_3_ is quite unknown yet, and the fact of disrupting the manure surface by tillage may support a rise in such emissions.

Dairy cattle barns are generally open, and naturally ventilated and the gas emission rate is dependent on several factors, such as thermal buoyancy forces, temperature, air humidity, and air pressure on the openings of the building [14,15]. Thus, choosing the right procedure to determine gas emissions in these systems is vital to obtain reliable information. Regarding ammonia emissions, one of the most commonly used methods is based on the barn input–output N-mass balance procedure [9]. This protocol relies on two assumptions: (i) the N excreted in the manure is equivalent to the N intake minus milk-N and (ii) differences between N excretion and manure N correspond to the irreversible-N losses by evaporation by default in the form of ammonia. Mass N balance is not an easy procedure mainly because N determinations in the feed, milk, and manure N are not free of error. Moreover, the procedure is restricted to N-containing gas emissions.

The use of tracer gases, either external (i.e., sulfur hexafluoride, SF_6_) [16,17,18] or internal, (i.e., carbon dioxide, CO_2_) [19], has been proposed as an alternative procedure, although the system relies on the assumption that a complete mixed air space in the building does exist and such a situation rarely exists in naturally ventilated buildings [20].

Other authors have measured CBP gas emissions directly by means of flux chambers [21], but in their case the impact and effect of the composting process by tilling on gas production and emission dynamics was completely overlooked.

Our study aims to analyze the impact of the CBP aeration performed by tilling over the gaseous emissions by differentiating phases within a CBP housing system; the procedure is based on the dynamic hoods protocol proposed initially by Seradj et al. (2018) [22]. Moreover, we determined the impact of long-term temperature dynamics on those emissions.

## 2. Materials and Methods

### 2.1. Barn Management

Measurements were performed in three selected commercial dairy cattle barns located in the surroundings of Lleida, inside the Ebro’s valley in the northeast of Spain; their specific characteristics are shown in Table 1. Barn selection was performed to seek representativeness, and the buildings were equipped with a loose housing system with a compost-bedded pack with a feed alley, cleaned mechanically at dawn and in the afternoon and surrounded by a retaining concrete wall designed to isolate the manure deposited into the feed alley from that deposited into the bedded pack without disturbing cow mobility. Milking parlors were placed at the center of the fully open buildings, which were naturally ventilated; no differential management was applied between the warm and cold seasons.

### 2.2. Animal Management

All farms raised Holstein Friesian cows, among 1 and 4 parturitions. The animals were artificially inseminated approximately 157 days after parturition and dried off 63 days before the anticipated next calving. Days in milk (DIM) were 83.6% [23]; during the dry period (16.4% of the days), the cows were managed in different facilities mixed with replacement heifers; however, no emission data from such a group was available in this trial. The cows were milked twice per day (milking times); meanwhile, the compost-bedded pack was being mechanically tilled.

Herd performance during both seasons is described in Table 2. Although the farms were selected to seek homogeneity, some differences between the farms persisted, such as herd farm size and manure accumulation (depth) due to variations in the manure emptying time protocol among the barns. The season did not alter the number of cattle, the average lactation number, the parturition interval, or the milk yield, but dry matter intake was slightly higher in the warm period (25.9 vs. 24.3, *p* = 0.04, SEM 0.21).

The cows received a total mixed ration (TMR) balanced according to the Agricultural and Food Research Council (1993) [24] with a minimum of 45% forage, including corn silage and alfalfa silage to support daily production of 30 to 35 kg of milk. Rations for the dairy cows were formulated to maintain an average CP (on a DM basis) between 16% and 17%. The main ingredients were soybean meal, corn, and barley (silage and grain). NDF, ADF, and NFC values varied between seasons, with higher winter proportions for NDF and ADF (345.3 vs. 320.4 and 209.5 vs. 188.4 g/kg DM, respectively) and the opposite was true for NFC during summer (387.1 vs. 365.4 g/kg DM). Altogether, the data regarding diet composition did not vary significantly between seasons and barns (*p* > 0.05).

The complete list of ingredients and chemical composition of the rations is presented in Table 3.

### 2.3. Environmental Parameters

During the experimental period, data on the temperature and wind speed were obtained from the two closest (less than 20 km) climatic control stations [Torres de Segre (41°51′90.9″ N 0°55′31.4″ E), close to barn 1, and El Poal (41°40′15.0″ N 0°52′37.7″ E), close to barns 2 and 3. The meteorological data show the geographical homogeneity of the farms located in the area (Table 2). The temperature was significantly different between the summer and winter seasons (36.4 ºC vs. 6.2 ºC *p* < 0.01) as well as the humidity (50.4% vs. 75.1%, *p* = 0.02). There were no significant differences between both climatic control stations except for the wind speed, where the climatic station of *El Poal* registered higher maximum wind rates than the *Torres de Segre* station (43 km/h vs. 16.7km/h, *p* < 0.01).

### 2.4. Sample and Data Collection

To assess total gaseous emissions, manure management of the CBP was differentiated into two distinct emission phases: static emission (SE), which occurs when the stored manure is not tilled upon directly, and dynamic emission (DE), which generates during and immediately after the mechanical tillage. Moreover, variations between the warm and cold seasons were performed in two sampling periods: winter (January–February) and summer (July–August, 2020). Sampling was performed at 1, 20, 40, and 60 days of each period.

#### 2.4.1. Static Emission

Four portable flow chambers (PFCs) made of PVC (20 cm Ø and 40 cm height) were designed for recollecting the emitted gas during the static phase. PFCs were placed on the surface of the composted bed. To avoid animal disturbance, their access to the PFCs was restricted by a 3 × 3 m perimeter fence. Two chambers were used to determine NH_3_ emissions, while the other two were used for CH_4_ collection (Figure 1). The chambers were interconnected two by two through a Teflon tube (4 mm Ø), and two distinct air fluxes were applied for adequate collection of each gas. Gaseous emission was calculated as the difference between the PFC in and outlet air concentrations fluxes (inlet air precedented from outside the barn). Air fluxes were generated by either a peristaltic pump (CH_4_) or an air pump (NH_3_) located in an outdoor analysis station.

#### 2.4.2. Dynamic Emission

To determine the emissions during the DE period, an airtight structure (ATS) was designed and placed on the surface of the compost bed (2m height × 1,5 m length × 2,5 m width). The ATS had a single air inlet and outlet placed in the low and high parts of the structure, respectively, and in the opposite walls (Figure 2). In the outlet structure, an air extractor (Soler and Palau TD-250/100 24-18W, 100 mm, Lleida, Spain) was placed, and air outlet flow was determined by an electronic anemometer (Extech SDL310: Thermo-Anemometer/Datalogger, Detroit, USA). Once the ATS structure was sealed, ventilation of the compartment started. When the air inside the simulator reached equilibrium (30 min was considered as a proper time-lapse) the ATS-isolated surface was mechanically tilled using a rototiller (Honda 1 speed Alpex, 5.5hp, 90cc, Satama, Japan) replicating the depth (25 cm depth) and tillage time performed on the rest of the barn by the rotary harrow. The total ventilation time lasted for 90 min: 30 min before and 60 min after tilling. During the ventilation period, air samples were taken continuously from both the inlet air (coming from outside the barn) and the outlet air flowing out from the air extractor placed on the ATS. To avoid over-estimation of CH_4_ and NH_3_ emissions due to rototiller emissions, the resultant gases proceeding from the fuel combustion of the rototiller were expelled from the ATS through a flexible plastic tube (PVC, 3 cm Ø) attached to the exhaust pipe of the rototiller. The air consumption of the rototiller was calculated from the technical indications of the engine and considered an additional out-flow way and included as such in the ATS model.

### 2.5. Air Sampling Protocol

#### 2.5.1. Methane

Air renewals (inlet plus outlet air) were continuously sampled through Teflon tubes (4 mm Ø) and conducted by a peristaltic pump (Gilson, Minipulse 3, Le Bel Villiers, France) at a flow rate of 10 mL/min (measured by an electronic gas flow meter Alltech, IL, USA) toward inert gas-tight bags (10 L volume, 15 μm thick) where the gas sampling was stored, following the protocol proposed by Morazán et al. (2013) [25].

After 24 h for SE and 1.5 h in the case of DE, airflow was stopped, and the air from the inside of the inert bags was sampled using hermetic syringes (Hamilton, NZ, USA) and immediately injected into 12 mL glass vials (model 039W, Labco, High Wycombe, UK) for further GHG analysis.

#### 2.5.2. Ammonia

Air renewals (inlet plus outlet air) were continuously sampled following the model proposed by Goldman and Jacobs (1953) [26] through Teflon tubes (4 mm Ø) using a vacuum air pump (KNF N035.3 AN.18-IP20, NJ, USA) at a 3 L/min (LZQ-1 0-5 LPM flow meter) flow rate while being bubbled into an acid solution (100 mL of H_2_SO_4_ 0.5 M) contained in glass impingers that trapped gaseous NH_3_ into aqueous NH_4_^+^ as shown in the following equation [27]:H_2_SO_4_ + 2NH_3_ → (NH_4_)_2_SO_4_

After the measurement time elapsed (24 h for SE and 1′5 h in the case of DE sampling), the acid solutions were sent to the laboratory for further NH_3_-N analysis.

### 2.6. Sample Analysis

#### 2.6.1. Methane

The samples were analyzed using a gas chromatograph 7890 A. The system was equipped with a flame ionization detector (FID) with a methanizer. An HP-Plot column (30 m long, 0.32 mm diameter) was used together with a 15 m-long pre-column. The injector and furnace temperatures were set at 50 ºC and 250 ºC, respectively. For the methanizer, the temperature was set at 375 ºC. Hydrogen was used as a carrier gas for the FID detector, with N_2_ as the compensatory gas, at 35 and 25 mL/min. The injected sample volume was 1 mL. Production of CH_4_ was then calculated according to Holland et al. (1999) [28].

#### 2.6.2. Ammonia

The ammonia nitrogen present in the medium was measured following the bases of the nitrogen total Kjeldhal (NTK) method [29] consisting of the addition of H_2_SO_4_ to transform the nitrogen into ammonia nitrogen. Our samples contained ammonium sulfate which was transformed to free NH_3_ once NaOH was added to the solution. Subsequently, free NH_3_ was determined by distillation and titration using boric acid (4% solution) and HCl (0.02N), respectively.

### 2.7. Emission Calculation

#### 2.7.1. Static-Phase Calculations

i.Methane

The CH_4_ concentration values (ppm) obtained from gas chromatography were transformed to mass/volume concentration (mg/m^3^) considering the molecular weight of each GHG, for which the ideal gas law was applied:$$C_{m}=\frac{Cv\times M\times P}{R\times T}$$
where *C_m_* is the mass/volume concentration (mg/m^3^), *C_v_* corresponds to the volume/volume concentration (ppm), *M* is the molecular weight of CH_4_, *P* is the atmospheric pressure, *R* is the universal gas constant, and *T* is the temperature in degrees Kelvin.

The 24 h CH_4_ (mg) emission was calculated as [outlet CH_4_ minus inlet CH_4_ concentration (mg/mL)] × air flow (mL) in each sampling period, assuming that the inlet air flow was equal to the induced outlet air flow (10 mL/min) for each pair of PFCs. CH_4_ emission was expressed as mg of CH_4_/m^2^ considering that the PFC surface was of 0.0628 m^2^

ii.Ammonia

The amount of NH_3_ trapped in the acid solution (mg NH_3_/L) was determined considering the N concentration (mg N/L) of the acid solution, the volume of the solution, and the molecular weight of NH_3_. The NH_3_ produced from the compost bed was calculated as the difference between the NH_3_ emissions of the inlet and outlet air. Daily NH_3_ emission (mg NH_3_/m^2^) was calculated as 24-NH_3_ harvested in the air traps from the outlet air divided by the PFC surface (0.0628 m^2^). The absolute amount of NH_3_ emitted per day was calculated as the product of NH_3_ concentration (mg NH_3_/L) and the airflow (3 L/min) recorded during each measurement (24 h for SP). Once corrected for the surface of PFCs, the results were expressed per unit area (mg NH_3_/m^2^).

#### 2.7.2. Dynamic Emission Calculations

i.Methane

The production of CH_4_ during DE was determined considering the airflow of the ATS and the concentration of CH_4_, which was corrected by the inlet air concentration. Moreover, the air consumed by the rototiller (3.8 L air/s, 5 min) during laboring was also considered and added to the air volume extracted from the ATS.

ii.Ammonia

Ammonia production was calculated as explained above for SE. The air renewal of the ATS (airflow; L/min) for 1 h (of the working air extractor) was determined considering the airspeed and the diameter of the extractor (10 cm) located above the ATS. The amount of NH_3_ captured in the acid traps solution (mg NH_3_/L) was determined as explained above for SE and the absolute amount of NH_3_ emitted from the ATS per hour was calculated considering the air renewal of the ATS per hour corrected for the air consumption of the rototiller.

### 2.8. Statistical Analysis

The data were analyzed with the Mixed Procedure in SAS (version 9.4; SAS Institute Inc., Cary, NC, USA). The barn was considered as the experimental unit for statistical purposes. The GHG data together with the NH_3_ emissions recorded in both phases during the two seasons of the study were analyzed as follows:Y_ijklm_ = µ + MP_i_ + S_j_ + (MP × S)_ijk_ + ε_ijkl_
where Y_ijklm_ is the dependent variable, µ is the overall mean, MP_i_ is the management phase (SE; DE), S_j_ is the seasonal effect (winter; summer), MP × S is the interaction effect among the previously described effects, and ε_ijkl_ is the error.

The Statistical significance and tendencies were declared at *p* ≤ 0.05 and 0.05 < *p* ≤ 0.10, respectively. For statistical analysis purposes, the methane data were previously normalized on a logarithm basis to minimize residuals.

## 3. Results

### Gaseous Emissions

The values registered for both managing phases are shown in Table 4; daily emissions are expressed per unit of surface (g/m^2^). Moreover, the integration of both phases, 23 h for SE plus 1 h for DE, allows the calculation of total emissions (g/m^2^ and day). Thus, the emissions of both CH_4_ and NH_3_ differed between both phases, with higher gas volatilization levels during DE (tilling period) than SE (CH_4_: 2.83 vs. 0.04 g/m^2^ and day, *p* < 0.01 and NH_3_: 2.21 vs. 0.76 g/m^2^ and day, *p* = 0.02). Moreover, as can be seen in Table 4, both phases showed higher gaseous emissions during the summer period, although the differences did not reach statistical significance. In any case and independently of the gas analyzed, the data presented in Table 4 are characterized by high variability. In the NH_3_ emissions, the variation coefficients were 18 and 60% for SE and 23 and 47% for DE during the winter and summer seasons, respectively. In CH_4_, the coefficients were 25 and 55% for SE and 23 and 44% for DE during the winter and summer seasons, respectively. On top of that, such variation was not constant but higher during the summer season.

The daily emissions from each barn [gas emitted by surface unit (g/m^2^) × total composted-bed surface] were expressed by cow (g gas/animal), milk production (g gas/kg milk), and N intake (g gas/g N intake), and the average values are presented in Table 5. Again, no significant differences were found between the seasons, but there was a tendency of increasing methane emissions during the summer period (*p* = 0.08 when expressed in (g/g N intake)).

## 4. Discussion

### 4.1. Methodological Approach

In bedded pack handling systems, two very different situations can be distinguished: (i) emissions produced when the bedded pack is composted conventionally by mechanical tilling and (ii) emissions in the bedded pack during resting when it is only submitted to cattle interactions. Differentiation between both emission situations among the CBP barns has never been considered so the tilling effect on polluting gas emissions was neglected. To solve this problem, two experimental phases were proposed (SE and DE). The authors are aware of the limitations of such an approach (i.e., the sampling areas were subjected neither to the cow’s interactions nor climatic incidences); however, the authors believe that such constraints are inevitable to simulate the activity over the bedded pack and fully harvest the gas emitted.

Three dairy cattle barns using the compost-bedded pack as housing systems were selected for the trial. The barns were located in the same area with identical animal genetics and production profiles. Even so, variations among the barns still existed, such as the number of animals per barn or the manure management conditions (i.e., storage times), and it is difficult to predict their impact on the experimental error.

To validate the methodology, two relevant gases were chosen: first, NH_3_ due to its importance as a final product of protein metabolism [30], and second, CH_4_ as a relevant end-product of bacterial carbohydrate metabolism [31,32].

### 4.2. Phase-Related Emissions

The results showed that, in CBP systems, the emissions during DE were higher than those registered during SE, for both NH_3_ and CH_4_ (*p* = 0.02 and *p* < 0.01, respectively). The authors are unaware of data that quantify gaseous CBP emissions from two distinct phases, although Wolf (2017) [33] found that CH_4_ emissions from CBP (as well as N_2_O and CO_2_) decreased suddenly after 20, 60, and 100 min after tilling (0.21, 0.013, and 0.082 g/m^2^ h^−1^, respectively), confirming that the processes that happen during tilling should not be neglected.

In fact, emissions for both gases mostly happen during tilling and this effect reached statistical significance. In the case of NH_3,_ mechanical tilling exposes the manure to air and dissolved CO_2_ is released faster. CO_2_ evaporation increases surface pH [34], and it shifts the balance NH_3_↔NH_4_+ toward NH_3_ and consequently increases N volatilization. The increase in CH_4_ emissions during tilling is more difficult to explain; mechanical aeration may break down anaerobic conditions [31,32] and consequently would decrease methanogen activity and release [35,36]. Probably, the release of CH_4_ accumulated under the upper manure layer by the action of the rotary harrow may compensate for the disruption in the methanogenesis process. Similar interaction was reported by Owen and Silver (2015) [37], when mixing and aerating solid manure piles increased CH_4_ emissions. After all, manure management in the CBP barns was not the same between both sampling phases, which leads to different behavior in gaseous dynamics as could be appreciated in the present study.

The picture we give, where measurement covers the full composting process, may be a better reflection of the real impact of CBP managing gas emissions. However, the authors are unaware of data from other authors where gas emissions during tilling have been considered; therefore, it is difficult to compare our findings against other papers, based additionally on the flux chamber protocol but measuring only the “static” situation.

These difficulties lie in: (i) the time aeration of the subtract is essential for the composting process and its effect on gas production must be relevant and (ii) gas emission measurements during the static phase should increase when the tilling process is experimentally omitted. In spite of that and for the present discussion, daily total emissions from other papers have been assumed as equivalent to the sum of both phases (DE and SE) defined in our assay.

### 4.3. CBP Emission: CH_4_

The methane emissions in the CPB averaged 1.0 and 4.75 g CH_4_/m^2^ day^−1^ (11.31 and 58.5 g CH_4_/cow day^−1^) in the winter and summer seasons, respectively; these results showed a high degree of unexplained variation that fitted well with information obtained from the available literature. In relation to the existing literature, following a similar approach, Leytem et al. (2011) [38] analyzed CH_4_ emission rates originating from three different areas (the out-barn pile area where manure was composted) reporting emissions that ranked from 2.6 to 35.2 g CH_4_/m^2^ day^−1^; van Dooren et al. (2011) [21]; using a similar flux chamber protocol determined a surface emission (g CH_4_/m^2^ day^−1^) of 14.4, 33,6 and 0,34 when the CBP was bedded with a peat and reed mixture, composted of wood and sand.

In terms of the emission rates of the cows (g CH_4_/cow day^−1^), our findings ranked from 11,31 to 58,5 and were similar to the values proposed by Owen and Silver (2015) [37] who obtained a narrower variation range between the seasons: 35.6 and 34 for summer and winter, respectively, whereas van Dooren et al. (2016) [39] in an intra-farm variation study ranked CH_4_ emissions as 24, 37.8, and 98.6 for the three different barns analyzed.

What is the relevance of manure emissions concerning enteric CH_4_ emissions? To address this question, we followed the meta-analysis carried out by de la Fuente et al. (2019) [40], where the enteric losses of CH_4_ in dairy cows ranged daily from 0.5 to 0.76 L/kg of live weight, considering an average of 700 kg in the farms analyzed [9]; in this scenario, manure CH_4_ emissions from CBP facilities may range from 3.6 (in winter) to 18.5 % (in summer). These results would confirm the relevance of manure management conditions in the total methane emissions in dairy cattle.

### 4.4. CBP Emission: NH_3_

Data related to irreversible NH_3_ losses coming from composting bed systems are scarce. Our results (total emissions of 1.86 and 4.08 g/m^2^ in the summer and winter seasons, respectively) fit reasonably well with those proposed by Leytem et al. (2011) [38], who performed monthly determination of ammonia in a dairy out-barn pile area where solid manure was composted and reported NH_3_ emission rates ranging from 0.34 to 3.45 g/m^2^. Van Dooren et al. (2011) [21], using a similar protocol also based on flux chambers, studied the emission of NH_3_ from different bedding materials (sand, composting wood chips, and a peat/reeds mixture). On average, the emissions from the surface (g/m^2^ d^−1^) of the pack were 9.96, 5.48, and 4.4 respectively. Moreover, a later study by this group [39] registered in four different CBP barns using different bedding materials reported the following values: 1.15, 3.70, 10.28, and 24.8 g/m^2^ d^−1^ for the four farms analyzed, which when expressed in daily emissions per cow account for 17.35, 46, 97.8, and 548 g/cow day^−1^. With caution related to specific differences in the used protocol and the potential effect of the bedding material employed, our registered emissions did not differ much from those obtained in the referred studies. Moreover, the results evidenced that even considering the large uncertainty due to the number of factors involved in the process, it is possible to provide a general estimate of NH_3_ emissions from dairy CBP facilities.

### 4.5. Season-related Emissions

Temperature has been previously related to NH_3_ and GHG emissions [9,41]. Our results confirmed a trend of higher emissions in the summer than in the winter season, despite not presenting statistical differences. The average temperatures recorded in both seasons were 5.3 ºC and 34.9 ºC, for the cold and warm seasons, respectively; this environmental temperature justify variations in the gas emissions between seasons. Our results would agree with those of several authors [41,42,43] who worked with the storage of dairy barn manure and/or slurry and described increased CH_4_ emissions in the summer season.

The link between environmental temperature and N-evaporation has been described in the existing literature although such a relationship can be modulated by two factors: first, CBP constitutes by itself an independent and very complex ecosystem where bacterial activity is the source of fermentation heat that might impact significantly on the media temperature [44]; thus, specific gradient variation between the environment and the internal CBP temperature has been consistently described [14]. Second, inside both the flow chamber and the ATS, ventilation rates are experimentally controlled and can hence interfere with the real barn surface ventilation. Both factors may have buffered the relationship between temperature and N-evaporation and hence lowered the theoretically expected emission rate.

## 5. Conclusions

Relevant amounts of polluting gases are generated during the composting process into the CBP system; when this measurement covers the full composting process, it reveals that most of the gas emissions occur when CBP is aerated mechanically by tilling. Thus, future studies conducted in CBP systems should include the specific impact of dynamic emissions. Moreover, our findings did confirm the significant impact of temperature on total gas emissions.

## Figures and Tables

**Figure 1 animals-13-01871-f001:**
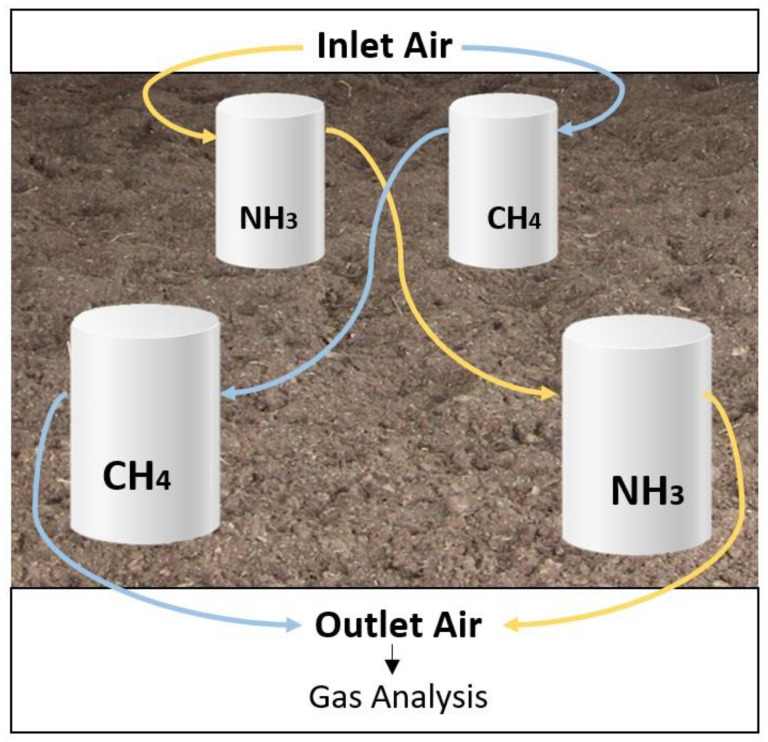
Portable flux chambers for CH_4_ and NH_3_ collection during the static emission.

**Figure 2 animals-13-01871-f002:**
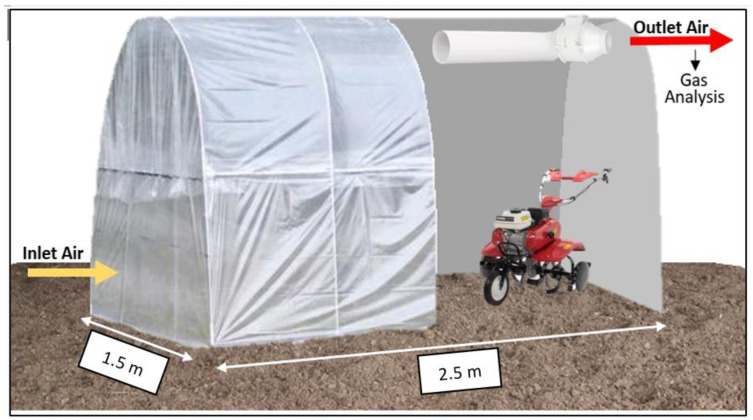
Airtight structure used for dynamic emission. A cut from the inside allows one to see the air extractor as well as the rototiller used to cultivate manure inside the structure.

**Table 1 animals-13-01871-t001:** Building characteristics, floor type, and manure-handing system in the free-stall dairy barns using composting beds as housing systems.

Barn^a^ nº	Geographic Coordinates	Floor Type and Manure Handing System	Length, m	Width, m	Feed Alley, m	Emission Surface, m^2^	m^2^ per cow
1	41°34′29.8″ N 0°27′07.0″ E	Daily bed cultivation, bed emptying up to 6 month, daily mechanical cleaning of the feed alley (2/day), and no bedding material	50	20.80 × 1 ^b^	4.0 × 1	1040	10.4
2	41°42′46.3″ N 0°46′44.7″ E	Daily bed cultivation, bed emptying up to 6 month, daily mechanical cleaning of the feed alley (3/day), and no bedding material	140	21.50 × 2	4.0 × 2	6020	12.4
3	41°42′33.5″ N 0°54′45.2″ E	Daily bed cultivation, bed emptying up to 6 month, daily mechanical cleaning of the feed alley (1/day), and no bedding material	84	11.45 × 3	5.0 × 3	2885	12.3

^a^ 1 = Ramaderia Fontanals; 2 = Cal Perches; 3 = Cal Padrí. ^b^ Number of equal pens per barn.

**Table 2 animals-13-01871-t002:** Cows’ performance in the compost-bedded pack housing systems under study (W, winter; S, summer).

Barn nº	Barn 1		Barn 2		Barn 3		Season		Barns	
Season	W	S	W	S	W	S	*p*-Value	SEM	*p*-Value	SEM
Climatic condition										
Temperature [ºC]	7.1	38.5	6.2	36.7	5.5	33.9	<0.01	1.00	0.9	15.06
Wind [km/h higher rate]	43.3	42.7	16.6	15.7	17.7	16.9	0.9	8.76	<0.01	0.38
Humidity [%]	71.0	40	81.7	53.2	72.7	58.2	0.02	4.49	0.8	12.85
Herd structure and performance										
Cows, nº	110	100	480	483	241	236	0.9	110.22	<0.01	3.34
Mean lactation, nº	2.3	2.2	2.2	2.2	2. 2	2. 2	0.4	0.02	0.5	0.03
Parturition interval, days	450	442	412	434	445	437	0.9	8.58	0.2	7.14
DMI [kg/day]	23.8	26.2	24.7	25.9	24.2	25.8	0.04	0.21	0.9	0.91
Milk yield [kg/day]	35.1	31.1	34.2	36.9	33.9	34.6	0.9	1.21	0.5	1.41

**Table 3 animals-13-01871-t003:** Dietary ingredients (kg of fresh matter, FM) and chemical composition (g/kg of DM unless otherwise noted) in different TMRs supplied to the compost-bedded pack (W, winter; S, summer).

Barn nº	Barn 1		Barn 2		Barn 3	
Ingredients [kg Fresh Matter/day]	W	S	W	S	W	S
Corn silage	21.00	22.00	12.00	26.00	14.00	25.00
Barley silage	5.00	6.88	12.00			9.37
Cottonseeds					2.00	1.65
Corn ears	8.02	7.24				
Alfalfa hay	2.50	2.00	3.00	3.70	4.00	
Barley straw	1.00	0.70	0.80	0.80	1.13	
Soybean hulls					1.03	1.00
Brewers grains	8.00	5.00	3.00	5.00		
Corn grain			5.00	3.50	7.15	3.18
Barley grain			2.50	2.50		
Soybean meal	2.46	2.14	3.40	3.50	2.87	3.73
Rapeseed meal	2.00	2.50	1.70	2.00	1.53	1.50
Bypass fat			0.30	0.30	0.24	0.35
Molasses	0.10	0.10	1.5			
Sodium bicarbonate	0.20	0.20	0.27	0.30	0.04	0.04
Palmitic acid	0.15	0.2	0.34	0.35		
Minerals and vitamins	0.35	0.35	0.47	0.47	0.43	0.55
Total fresh matter	48.28	47.11	46.00	48.10	34.42	46.72
Chemical composition [g/kg DM]						
Dry matter [g/kg fresh matter]	480.9	488.2	525.6	535.0	668.4	496.0
CP	165.0	165.0	170.1	169.8	170.0	168.1
NDF	384.3	334.8	321.3	313.2	330.3	313.3
ADF	215.8	198.2	197.8	187.2	215.0	180.0
NFC	331.6	378.7	384.5	390.9	380.1	391.7

**Table 4 animals-13-01871-t004:** Results for phase and seasonal emissions for the gases of interest, expressed per surface units.

Gaseous Emission (g m^−2^ Day^−1^)	SE	DE	Total Emission	Phase Effect	
				SEM	*p*-Value
CH_4_					
Winter	0.004	0.99	1.00	0.09	<0.001
Summer	0.09	4.65	4.74	0.283	0.03
NH_3_					
Winter	0.39	1.46	1.86	0.075	0.01
Summer	1.13	2.95	4.08	0.233	0.30

**Table 5 animals-13-01871-t005:** Emission of both gases of interest expressed per animal, kg milk, and g N intake for both seasonal periods.

Compost-Bedded Pack Emission [g Animal^−1^ Day^−1^]	Gaseous Emission		*SEM*	
	NH_3_	CH_4_	NH_3_	CH_4_
Winter	20.9	11.31	4.13	2.85
Summer	50.2	58.5	22.2	27.0
*SEM*	22.56	27.15	-	-
*p*-Value	0.57	0.14	-	-
[g milk^−1^ day^−1^]				
Winter	0.61	0.33	0.12	0.08
Summer	1.41	1.64	0.61	0.73
*SEM*	0.62	0.74	-	-
*p*-Value	0.44	0.08	-	-
Compost Bed Pack [g g N intake^−1^ day^−1^]				
Winter	0.03	0.01	0.006	0.004
Summer	0.08	0.09	0.03	0.04
*SEM*	0.03	0.04	-	-
*p*-Value	0.61	0.12	-	-

## Data Availability

Not applicable.

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
