# Peer review of "Measurement of Methane and Ammonia Emissions from Compost-Bedded Pack Systems in Dairy Barns: Tilling Effect and Seasonal Variations"

_animals, 2023, doi:10.3390/ani13111871_

Round 1
Reviewer 1 Report
The article raises a significant problem regarding cattle breeding.
Unfortunately, it is inconsistent and requires improvement.
The most important problem is the omission of statistical analysis in the application.
The discussion is a review of literature. Lack of connection obtained with this literature. Mixing chapters: Materials and Methods, Results and Discussion.
Below are additional detailed tips:
Page 1, line 22 - Name abbreviations should not be used in the abstract.
Page 1, line 26-unit written incorrectly instead of "/m2" I suggest "m-2"
Page 23, line 101 - The tables in the text start with number 2 - the order of quotation should be maintained.
Page 3 line 116-155-it would be clearer if the installation diagrams were posted.
Page 6 line 256 - 3.1 Barn Management and Herd Performance - transfer to the methodology
Page 8 line 280 - 3.2 Composition - transfer to the methodology
Page 8 line 293 - 3.3 Environmental Varights - transfer to the methodology
Page 9 Line 309-310-"Both Phase 309 showed higher gaseous emissions in the summer, although Differences did not reach statistical significance" - incorrect formation. There were no differences from a scientific point of view. Similar problem (page 9, line 310). For this purpose, an external statistical analysis is carried out.
Page 12, line 461-462 Our 461 results confirmed that gas emissions were higher in the summer than in the winter season 462. - Not true - no significant statistical differences.
Reviewer 2 Report
the authors raise important aspects of ammonia and methane emissions in the publication. The publication requires some additions. Comments are included in the publication

Reviewer 3 Report
3 you do not have seasonal variations only. Also SE and DE. They seem more interesting, the seasonal effect is already known.
12 manure emissions?? Please rewrite the sentence
19 only manure or also manure? Please rewrite
41 60% where? In Spain? It may vary between the countries.
42 Sketch the stable and mark the measuring points
56/57 alternative to the loose-hous- 56 ing systems with cubicles?
60/75 Please also include in this section the influence of air temperature on NH3 and CH4, citing the meta-analysis that confirms the temperature effect on both gases. [https://www.sciencedirect.com/science/article/pii/S1537511018307633 -- Effects of housing system, floor type and temperature on ammonia and methane emissions from dairy farming: A meta-analysis Jernej Poteko, Michael Zähner, Sabine Schrade]
71 abb this study with SF6 [https://www.sciencedirect.com/science/article/abs/pii/S1352231018300712]
77/78 The objective of the study is inadequately described and requires revision.
89/90 write the manure removal times
118 January-February Year?
155 provide a photo of sampling in the barn
156 provide a scheme of sampling
274 Table 2……P should be P-Value; make the significant p-values bold
275 You should use [ ] brackets for units in all tables. E.g. [°C]
275 Layout of table 2 can be optimised (e.g. with less rows)
254-292 Chapters 3.1 and 3.2 incl. tables should be a part of Material and methods and not results
291 the statistical analysis of chemical composition of diet ingredients may be useful to show differences between barns and seasons. Did you try the diet as a factor in your model?
291 Kg -> kg Make it correct in whole manuscript.
294 you can avoid the first sentence.
304 you can avoid the first sentence.
304-305 daily emissions are expressed per unit of surface (g/m2 ) à this is information for Matherial and methods chapter.
307-308 in winter or summer?
319 Compost Bed Pack (g/m2 & day) ----write the unit somewhere else, e.g. near emission. Compost Bed Pack has no unit.
319 divide the results on NH3 and CH4, and write the emission and SEM of each gas together.
319 Did you try to make a stat. analysis for each barn? Perhaps the variation would be lower?
245&318&338 in tables 4 and 5 you should compare or analyse the emission for each barn separately. Did you try the barn as a factor in your model? What was the p-value of barn-effect?
253 you have no difference between emissions in winter and summer. Did you try to make a stat. analysis with the emission data of just 23 hours (SE) separately? What is the p value? The variance of 23h (SE) and 1h (DE) may be interesting? The better way to present your results would be to divide the results on SE and DE, separately for each barn.
Overall, you shuld rething the main messege of this paper. Is it the measuring of emissions SE and DE?
Rethink the structure of results and fokus on SE and DE.
373 p value?
462 p value?
Round 2
Reviewer 3 Report
1 change title: Measurement of methane (CH4) and ammonia (NH3) emissions from compost-bedded pack in dairy barns: tilling effect and seasonal variations è Measurement of methane and ammonia emissions from compost-bedded pack in dairy barns: tilling effect and seasonal variations
Author Response
Dear Reviewer,
Thank you for your suggestion, we agreed with your point of view and we have already changed the title to the one you recommended in the new version of the paper.
Many thanks for your help